# Physical Activity Behaviors and Physical Work Capacity in University Students during the COVID-19 Pandemic

**DOI:** 10.3390/ijerph19020891

**Published:** 2022-01-14

**Authors:** Grzegorz Bielec, Aneta Omelan

**Affiliations:** 1Faculty of Physical Education, Gdansk University of Physical Education and Sport, 80-336 Gdansk, Poland; 2Department of Tourism, Recreation and Ecology, Faculty of Geoengineering, University of Warmia and Mazury in Olsztyn, 10-719 Olsztyn, Poland; aneta.omelan@uwm.edu.pl

**Keywords:** physical activity, body composition, PWC170, COVID-19, students

## Abstract

*Objective*. The COVID-19 pandemic led to restricted access to sports and recreation facilities, resulting in a general decrease in physical activity. Many studies present the results of on-line questionnaires conducted during the pandemic, but there are few reports of objectively measured indicators of physical condition. Thus, the objective of this study was to assess the changes in physical work capacity, body composition, and physical activity behaviors in university students during 14 weeks of lockdown. *Material and Methods*. Twenty students of Tourism and Recreation (13 female and 7 male) participated in the study. The first examination was conducted in November 2020, and the second in March 2021. Body composition was assessed with a Tanita 418 MA device. The students performed the PWC 170 cycling test and completed the International Physical Activity Questionnaire (short version) on-line. *Results*. Neither physical work capacity nor body composition parameters changed substantially during the analyzed period. In the female students, vigorous physical activity decreased significantly, but no substantial changes occurred in weekly metabolic equivalent of task. In male students, walking days and metabolic equivalent of task decreased, but the changes were not significant. *Conclusions*. Fourteen weeks of COVID-19 lockdown had little effect on the body composition, physical work capacity level, and physical activity habits of Tourism and Recreation students. Studies with larger groups of participants should verify the current conclusions, and care should be taken when extrapolating to other populations.

## 1. Introduction

The benefits of physical activity and its effects on human health and well-being have been well researched by scientists and extensively documented [1,2,3,4,5,6]. A lack of physical activity is a risk factor for many diseases, which is of great importance from a public health point of view [7,8,9,10]. The global cost of physical inactivity in 2013 was estimated to be 54 billion international dollars (INT$) per year in direct health care, with an additional INT$ 14 billion in lost productivity. Inactivity accounts for 1–3% of national health care costs, although this excludes costs associated with mental health and musculoskeletal conditions [11].

In accordance with WHO recommendations, adults aged 18–64 years should do at least 150–300 min of moderate intensity aerobic physical activity per week, or at least 75–150 min of vigorous intensity aerobic physical activity, or an equivalent combination of moderate and vigorous intensity activity [11]. However, despite the promotion of physical activity and widespread access to knowledge about healthy lifestyles, 1 in 4 adults worldwide currently do not meet the global physical activity recommendations set by the WHO [12].

Adults also include university students, who can be considered a special social group. The transition from secondary to tertiary education is considered critical because this is often when, from a public health perspective, young people’s behavior and lifestyle change dramatically for the worse [13]. This change is associated with, among other things, poor eating habits and a reduction in the amount of physical activity performed by young people [14,15]. Unfortunately, the physical activity level of young people studying in higher education is too low [16,17].

The coronavirus pandemic has affected the physical activity of people all over the world. During the first wave, sports and recreational facilities were closed in many countries (e.g., swimming pools, gyms, fitness clubs, playgrounds) and organized physical activities were canceled. There were also limited opportunities for outdoor sports and recreation. For example, during the first lockdown in Poland, golf courses and hiking trails could not be used, and forest areas were excluded from recreational and tourist use, further limiting opportunities to be physically active. During the second wave of the pandemic, regulations were no longer as stringent, but many restrictions were maintained. Before the coronavirus pandemic, 65% of Poles declared that they were physically active at least once a month, and of these, 39% followed World Health Organization recommendations [18]. During the first lockdown, the number of physically active people fell by 4%, and during the second lockdown, it increased by 2% (to 63%) [19]. Similarly, research conducted around the world indicates that people’s physical activity behavior changed over this period, and a large amount of recent research shows that, while most people decreased their involvement in physical activity, some increased it [20,21,22]. The situation of enforced confinement of millions of people to their homes and its impact on health in general, which has never been seen before in the modern world, will probably be analyzed by researchers for many years to come. However, many are already drawing attention to the particular importance of exercise in maintaining physical and mental health during lockdown, as well as mitigating the course and effects of COVID-19 [23,24]. Physical activity during lockdown/quarantine has also been encouraged by the World Health Organization, which has maintained its recommendations for physical activity [12].

The coronavirus pandemic was also not without its effects on the lifestyle of university students. The consequences of lockdowns included closure of universities, a shift to e-learning, and the closure of dormitories, which forced many students to return to their family homes. Like other citizens, students were deprived of access to sports and recreational facilities, but they were left with the option of continuing their physical activity at home, and, in many places, also outdoors. Therefore, from a research perspective, it was of interest whether this group of young adults took advantage of the limited opportunities to be physically active, and if so, to what extent. In our research, we focused on a group that (it is assumed) should be more active than most students in other fields of study: Tourism and Recreation students. This assumption reflects the fact that they have chosen a course of study in which physical recreation is an important part of education. The university curriculum includes many subjects (theoretical and practical) that prepare these students to be specialists in active forms of leisure and promoters of physical activity. Therefore, it is interesting to see whether they confirm the special importance of physical exercise for health (physical, mental, and social) by their own positive attitude towards it.

Thus, the objective of the study was to assess the changes in (i) physical work capacity, (ii) physical activity levels, and (iii) body composition in Polish university students during the so-called second wave of the COVID-19 pandemic.

## 2. Material and Methods

### 2.1. Design and Participants

The study began in November 2020, two weeks after the Polish government’s decision to institute restrictions on public life due to the increase in the number of COVID-19 infections. University education was conducted exclusively by internet.

The targeted group consisted entirely of third-year students of Tourism and Recreation at the University of Warmia and Mazury in Olsztyn. The inclusion criteria included (i) a self-reported good state of health, (ii) no prior COVID-19 infection, and (iii) the ability to participate in laboratory examinations on the university campus. An invitation was sent to 67 Tourism and Recreation students using the Microsoft Office Teams communicator. All procedures were described in this invitation. Additionally, students were surveyed on-line about their personal experience with COVID-19. The students were asked to state if they had ever been infected with the SARS-CoV-2 virus or had been under quarantine because of co-residents’ illness. Three students declared recovery from COVID-19 in the past four months and they were excluded from the study. Initially, 28 students who met the inclusion criteria agreed to participate in the study. Four students withdrew their participation shortly after this, and then four students decided not to participate for health or personal reasons. Thus, 13 female and 7 male students aged 21–24 years (mean 22.6 years) took part in the study. The anthropomorphic characteristics of the examined group are presented in Table 1. The students were asked not to change their eating habits during the study.

All of the procedures described below were conducted twice: in November 2020 and in February/March 2021. The period between the first and the second examination of each participant was 14 weeks ± 2 days. The participants were asked to refrain from alcohol, caffeine, energy drinks, and strenuous physical activity for 24 h prior to examination.

All measurements were conducted following appropriate procedures for safety during the pandemic. The measuring tools were treated with 70% alcohol before each measurement. The researchers and participants wore face masks (except for the PWC 170 test) and kept an appropriate distance from each other.

### 2.2. Data Collection

#### 2.2.1. International Physical Activity Questionnaire

To assess the students’ level of physical activity, the International Physical Activity Questionnaire (IPAQ) was employed. The short version of the questionnaire contains seven questions regarding the different kinds of physical activity performed in the last seven days. The respondent indicates how many days and how much time he/she devoted to vigorous physical activity, moderate physical activity, and walking. The metabolic equivalent of task (MET) is calculated based on the participant’s responses to the questionnaire. The metabolic cost of vigorous activity is considered to be 8 MET/min; that of moderate activity, 4 MET/min; and that of brisk walking, 3.3 MET/min [25]. The respondents are then classified into three groups on the basis of their physical activity level [26]:Low physical activity—total physical activity of less than 600 MET/week;Moderate physical activity—total physical activity of more than 600 MET/week or vigorous physical activity of more than 480 MET/week;High physical activity—total physical activity of more than 3000 MET/week or vigorous physical activity of more than 1500 MET/week.

The questionnaire was sent to the participants via Microsoft Teams the day before examination in a laboratory. The students completed the questionnaire on-line and sent it back the same day.

#### 2.2.2. Anthropomorphic Measurements

Anthropomorphic measurements were taken in the morning. The participants wore light sports clothes and were barefoot. Body height was determined to the nearest 0.1 cm with a Seca 216 stadiometer (Seca GmbH, Hamburg, Germany). Body composition was estimated using a Tanita BC 418 MA analyzer (Tanita Corp., Tokyo, Japan). Body composition measurements included body mass, body mass index, muscle mass, percentage of water, percentage of adipose tissue, visceral tissue, and basal metabolic rate. The measurements were taken twice, and the coefficient of variation was calculated for each pair of measurements: it ranged from 2.1 to 3.1.

#### 2.2.3. Physical Work Capacity 170 Test

The Physical Work Capacity 170 (PWC 170) test was used to estimate the subjects’ work capacity at a heartrate of 170 beats per minute. The test was conducted on a Monark 874-E (Monark, Vansbro, Sweden) cycloergometer just after completing anthropomorphic measurements. Each participant adjusted the saddle of the cycloergometer to his/her body height. A Polar H10 heartrate monitor (Polar Electro, Kempele, Finland) was worn by the participants during the testing procedure. The test started with a one-minute cycling warm-up with a load of 30 W. Then the participant cycled constantly for 5 min with a load of 1 W per 1 kg of body mass. After one minute of rest, the participant cycled for another 5 min with a load of 1.5 W per 1 kg of body mass. The test finished after a one-minute cycling warm-down with a load of 30 W.

The result of the PWC 170 test was calculated using the following formula:PWC170 = P1 + (P2 − P1) × (170 − HR1)/(HR2 − HR1)
where P1 is the power (load) of the first effort, P2 is the power of the second effort, HR1 is heart rate during the first effort, and HR2 is heart rate during the second effort [27].

### 2.3. Statistics

Statistical calculations were performed using Statistica 12.0 (StatSoft, Tulsa, OK, USA). The Shapiro–Wilk test did not find significant deviations from normality. The results of the first (November 2020) and second (February/March 2021) measurements were compared using *t*-tests. Statistical significance was set at *p* < 0.05.

## 3. Results

Table 1 presents changes in the students’ measurements over the 14-week period. The mean body mass of the male students increased slightly, and the basal metabolic rate of both female and male students decreased. However, the changes in these variables were not statistically significant. Interestingly, the values of body mass, body mass index, body fat, water content, and muscle mass remained almost unchanged in the female students.

Table 2 presents the results of the Physical Work Capacity 170 (PWC170) test. The work capacity, expressed in watts [W], increased slightly in both the female and male students, but the changes were not significant. The work capacity expressed in watts per kilogram of body mass [W/kg] did not change significantly in the analyzed period, either.

Table 3 presents the changes in the students’ physical activity (PA) levels based on their responses to the International Physical Activity Questionnaire (IPAQ). The time that female students devoted to vigorous physical activity decreased substantially. However, the time they spent on moderate physical activity, their walking time, and the energy they expended increased. In contrast, the male students increased the time they devoted to vigorous physical activity, although the change was not significant. The time they spent on moderate physical activity and their energy expenditure decreased. Overall, the smallest changes occurred in the number of days devoted to moderate physical activity, and in the male students, this number remained unchanged. Additionally, the female students did not change the number of days devoted to walking.

## 4. Discussion

The objective of this study was to assess the potential changes in physical work capacity, body composition, and physical activity levels among Tourism and Recreation students during the COVID-19 pandemic. We hypothesized that restricted access to sports and recreation facilities would cause a decrease in their physical activity levels and physical work capacity. Our results indicate that, during the 14 weeks of lockdown in which the study took place, the students’ metabolic equivalent of task (MET) changed very little. The MET of the female students slightly increased, whereas that of the male students slightly decreased, but neither of these changes was statistically significant. We did not examine the physical activity of the students before the pandemic; therefore, we cannot assess whether the PA of the students declined during the lockdown or not.

When comparing these results to those of other studies, it is important to remember the contexts in which the studies took place. In Poland, during the second wave of the COVID-19 pandemic, access to sports facilities, i.e., to swimming pools, sports halls, and fitness clubs, was limited only to competitive athletes. Moreover, typical winter outdoor recreation facilities like skate rinks and ski slopes were closed in accordance with government regulations. Thus, the effect of pandemic-related restrictions on people’s physical activity may have depended on regional circumstances. For example, Sanudo et al. [28] reported that the PA levels of Spanish students decreased markedly during the first wave of the pandemic. However, it should be remembered that in Poland, and many other places, restrictions on public life were much more severe during the first wave than during the following waves of the pandemic. In Poland, during spring 2020, for example, going out of home was only allowed when visiting food shops, pharmacies, or commuting to work. It was not possible to participate in any outdoor or indoor sports or recreational activities because of the closed facilities. During the first wave of the pandemic, in Hungary, Acs et al. [29] did not detect substantial changes in the amounts of vigorous and moderate PA by Hungarian students surveyed online with IPAQ before and during the lockdown, but their weekly walking time decreased markedly. Chinese students also reported on-line that their level of PA decreased during the first wave of the pandemic in spring 2020, and more than 50% of those students did not meet the recommendations of the World Health Organization concerning the amount of physical activity [30]. As for Italian university students, there were significant decreases in their amounts of vigorous PA, moderate PA, and particularly in their amount of walking time during the pandemic [31]. A study of Italian medical students (female and male) also found substantially lower levels of PA during the lockdown than before it [32]. Interestingly, the mean total MET value of PA performed by the Italian medical students before the COVID-19 pandemic was almost equal to the mean total MET value of PA performed by the Polish Tourism and Recreation students during lockdown (1588 and 1538, respectively). The authors of the abovementioned Italian studies found that high levels of physical activity by the students before the pandemic were associated with higher physical activity levels during lockdown. However, Maltagliati et al. [33] found that subjects with previously strong PA habits demonstrated very low PA levels during the lockdown. Similarly, American students who were most physically active before the pandemic decreased their levels of PA substantially during lockdown [34]. Based on reports in the literature, we speculate that the Tourism and Recreation students in our study might have reduced their PA levels after the COVID-19 pandemic began. Some studies have found that, before lockdown, Polish students in pro-health fields of study (e.g., physiotherapy, physical education, tourism and recreation) declared high levels of PA (2000–10,000 MET/week). The abovementioned studies were conducted with the IPAQ short-version scale, which means that the results are comparable with ours [35,36].

In our study, the 14-week period of the pandemic did not cause substantial changes in the physical work capacity of the Tourism and Recreation students. In both examinations, the female students’ work capacity was classified as low, whereas that of the males was classified as average [37]. Kapilevich et al. [38] obtained similar results in the PWC170 test with male sports science students with low and average levels of extra-curricular physical activity. Studies conducted before the pandemic also reported similar values on the PWC 170 test for college-age women and men [39,40]. The lack of substantial changes in the work capacity of the examined students could be explained by their non-athlete status. Regular physical training raises the level of physical work capacity, but that level remains constant in non-athletes [41]. Moreover, non-athlete students and non-endurance athlete students usually display a lower level of physical work capacity than endurance sport students [42].

Our study did not find that the body composition of the Tourism and Recreation students changed significantly. The lack of substantial changes in their body composition may be due to two factors. First, their physical activity behaviors did not change during the study period. Second, our study was conducted for a relatively short period of time (14 weeks), and changes in their body composition may have been noticeable over a longer period. For example, Chwalczynska and Andrzejewski [43] reported significant changes in body composition between December 2019, i.e., four months before lockdown, and February 2021, during the so-called third wave of the pandemic. Body mass and body mass index increased in their male students, whereas fat mass increased in their female students. Similarly, Pop and Ciomag [44] took anthropometric measurements in Romanian students in spring 2018 and in December 2020, i.e., during the second wave of the pandemic, and found that their body mass index increased significantly in the analyzed period.

We realize that the number of participants is a limitation of our study. This number is due to the study being conducted during a period of remote learning without frequent direct contact with the students. Most of our Tourism and Recreation students lived outside our university town, and the student dormitories were closed. Therefore, some students did not accept our invitation to participate in the experiment because they did not have a place to stay. However, direct contact with the smaller group of students let us take objective measurements instead of only relying on subjective declarations of physical work capacity and body composition. This is a strength of our study, as such measurements were not common during that stage of the pandemic. Additionally, although the validity of the IPAQ short version examination in terms of metabolic equivalent of task (MET) determination is moderate, this questionnaire is used by national health institutions to assess the physical activity levels of large cohorts [45]. In any case, many of the results presented here should not be generalized because of local differences in sanitary restrictions. Instead, these results and those of similar studies provide an interesting basis for comparison.

## 5. Conclusions

The restricted access to sports and recreational activities during 14 weeks of the COVID-19 lockdown did not substantially influence the physical activity behaviors of Tourism and Recreation students. The physical work capacity of both the male and female students did not change significantly during this time, and no effect on their body composition was apparent. These conclusions should be confirmed with studies involving a greater number of participants and taking place in various locations.

## Figures and Tables

**Table 1 ijerph-19-00891-t001:** Measurements of Tourism and Recreation students during a 14-week period of the COVID-19 pandemic. Data is presented as mean value ± standard deviation.

	Body Height [cm]	Body Mass [kg]		Body Mass Index [kg/m^2^]		Body Fat [%]		Water [%]		Muscle Mass [kg]		Basal Metabolic Rate [kcal]		Visceral Tissue [Level]	
		Pre	Post	Pre	Post	Pre	Post	Pre	Post	Pre	Post	Pre	Post	Pre	Post
Females (*n* = 13)	166.3 ± 5.5	58.5 ± 9.7	58.6 ± 10.2	21.0 ± 2.6	21.0 ± 2.7	24.1 ± 6.6	24.1 ± 7.3	56.2 ± 5.0	56.3 ± 5.5	41.7 ± 4.6	41.6 ± 4.9	1360.6 ± 142.0	1351.8 ± 145.5	1.3 ± 0.6	1.4 ± 0.8
*t*-test *p* value		0.85		0.86		0.87		0.70		0.71		0.23		0.16	
Males (*n* = 7)	176.7 ± 7.9	69.0 ± 8.6	69.9 ± 9.4	22.1 ± 2.3	22.3 ± 2.5	17.2 ± 5.6	16.8 ± 5.7	59.1 ± 4.8	58.6 ± 5.4	53.5 ± 5.9	54.8 ± 6.0	1754.4 ± 166.9	1739.7 ± 167.3	2.1 ± 1.5	2.1 ± 1.8
*t*-test *p* value		0.06		0.18		0.80		0.55		0.14		0.15		1.0	

**Table 2 ijerph-19-00891-t002:** Physical Work Capacity 170 test with Tourism and Recreation students during a 14-week period of the COVID-19 pandemic. Data are presented as mean value ± standard deviation.

	PWC170 [W]		PWC170 [W/kg]	
	Pre	Post	Pre	Post
Females (*n* = 13)	111.7 ± 26.1	118.0 ± 43.7	1.9 ± 0.3	2.0 ± 0.8
*t*-test *p* value	0.50		0.39	
Males (*n* = 7)	185.8 ± 69.7	186.9 ± 53.1	2.6 ± 1.0	2.6 ± 0.7
*t*-test *p* value	0.95		0.87	

**Table 3 ijerph-19-00891-t003:** Physical activity levels in Tourism and Recreation students in a 14-week period of the COVID-19 pandemic. Data are presented as mean value ± standard deviation.

	Vigorous PA [Days]		Vigorous PA [min/day]		Moderate PA [Days]		Moderate PA [min/day]		Walking [Days]		Walking [min/day]		Total Metabolic Equivalent of Task [MET/week]	
	Pre	Post	Pre	Post	Pre	Post	Pre	Post	Pre	Post	Pre	Post	Pre	Post
Females (*n* = 13)	1.6 ± 1.3	1.1 ± 1.2	20.3 ± 17.6	16.1 ± 18.2	2.9 ± 1.1	3.0 ± 0.9	26.1 ± 16.2	30.7 ± 16.0	7	7	26.1 ± 8.9	35.3 ± 18.0	1370.3 ± 770.4	1486.9 ± 843.0
*t*-test *p* value	0.02		0.06		0.33		0.10		1.0		0.06		0.33	
Males (*n* = 7)	0.71 ± 1.1	0.85 ± 1.4	12.85 ± 17.0	17.14 ± 24.2	2.8 ± 1.5	2.8 ± 1.5	34.2 ± 26.9	25.7 ± 16.4	7	6.2 ± 1.2	45.7 ± 39.2	47.1 ± 22.1	1701.7 ± 1050.0	1633.5 ± 861.6
*t*-test *p* value	0.60		0.50		1.0		0.46		0.18		0.89		0.85	

## Data Availability

The data presented in this study are available on request from the corresponding author.

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
