# Peer review of "Physical Activity Behaviors and Physical Work Capacity in University Students during the COVID-19 Pandemic"

_ijerph, 2022, doi:10.3390/ijerph19020891_

Round 1
Reviewer 1 Report
Thank you for the opportunity to review the work under the title Physical activity behaviors and physical work capacity in university students during the COVID-19 pandemic.
The article has the correct structure. The conclusions correspond to the results, however, it should be noted that the studies were conducted on a small and specific group. It is worth emphasizing that these results should not be related to the population. It should be assumed that Tourism and Recreation Students are active and athletic. This is indicated by their area of interest and the chosen field of study. The population lockdown has had a disastrous effect on activity levels and weight gain in other social groups, such as children and adolescents.
Author Response
We would like to thank you for objective and thorough review. Proper comments were taken into account and there were contribute to improve the scientific level of the paper.
The article has the correct structure. The conclusions correspond to the results, however, it should be noted that the studies were conducted on a small and specific group. It is worth emphasizing that these results should not be related to the population.
We agree that the results cannot be representative of the entire students population, but they can be representative of the students population of the Tourism and Recreation Faculty
It should be assumed that Tourism and Recreation Students are active and athletic. This is indicated by their area of interest and the chosen field of study.
We are aware that students of Tourism and Recreation can be considered a specific group, as we emphasized in the introduction (80-85). And that is why this group appeared more interesting to us.
Reviewer 2 Report
Dear authors,
For a descriptive pre-post study the sample is quite low. Thus, the results lack validity. The low sample may have influence the significance of the results.
It is also not clear why was that sample selected and why that low response rate was obtained.
The methods also needs more information and structure to clearly state the next things and make it more clear for the readers: a)study design, b)study sample, c)variables, d)data collection process, e)ethical aspects, f) data analysis
No information about the sample characteristics is included in the results.
The results need more written information to introduce the tables.
This sentence in the conclusion is not supported by the results. Your results are not significant "The physical work capacity of both the male and the females students increased slightly during this time"
Kind regards
Author Response
We would like to thank you for objective and thorough review. Proper comments were taken into account and there were contribute to improve the scientific level of the paper.
For a descriptive pre-post study the sample is quite low. Thus, the results lack validity. The low sample may have influence the significance of the results.
We agree with this comment and this issue is explained in the last paragraph of "Discussion" section.
It is also not clear why was that sample selected and why that low response rate was obtained.
We provided more information about recruitment for our study in "Material and Methods" section. The explanation about low response rate is also presented in the last paragraph of "Discussion" section
The methods also needs more information and structure to clearly state the next things and make it more clear for the readers: a)study design, b)study sample, c)variables, d)data collection process, e)ethical aspects, f) data analysis
We did some revisions in the structure of "Material and Methods" section. However, the order of sub-section was created based on IJERPH examples. For example "Ethical clearance" section (Institutional Review Board Statement and Informed Consent Statement) always appears after "Discussion" section in IJERPH articles.
No information about the sample characteristics is included in the results.
Anthropomorphic characteristics of the sample is presented in Table 1 in "Results" section.
The results need more written information to introduce the tables.
More information introducing the tables were added.
This sentence in the conclusion is not supported by the results. Your results are not significant "The physical work capacity of both the male and the females students increased slightly during this time"
This sentence reflects the results of physical work capacity (PWC) expressed in watts that is presented in Table 2. This value increased from 111.7 W to 118.0 W in women, and increased from 185.8 W to 186.9 W in men. Therefore we wrote that PWC increased slightly.
Reviewer 3 Report
40-46: I suggest removing the fragment because it adds little to the substantive part of the manuscript and only makes it less readable
74-77: Given the characteristics of youth……of forced isolation- It’s an assumption which should be removed to the Discussion section or if it is a fact that can be confirmed by the literature on the subject so an adequate reference is needed.
79-84: In our research we focused on a group particularly predisposed to being physically active: …. towards it. As above- It’s only an assumption that should be removed to the Discussion section or if it is a fact that can be confirmed by the literature on the subject so the adequate reference is needed. Because not every student nor most of them of this field may be interested in being physically active, but only looking for a pleasant job and the opportunity to spend time/work in beautiful and interesting corners of the world, as an animator, not necessarily from physical group activities, but rather as a guide for excursions, for example.
89-96: it is not a Designed of the study, but rather a statement of the fact about the situation in the country during the pandemic period. So it is a suggestion to move this part to the Introduction. The section Design of the Study should contain chapter-specific information.
103-104: All Participants should be described in a more precise way, i.e. height, body weight, BMI. Moreover, in the Materials and Methods section, information about the health of the study participants is stored. Did they suffer from COVID-19, whether they were before or after vaccination, whether they suffered from other ailments, whether they used any specific dietary intervention, whether their condition and diet were similar, whether they supplemented products with an ergogenic effect or before the PVC 170 test? have used CNS stimulants such as caffeine. All these factors affect the results of stress tests, as well as mental well-being and the results of the questionnaire tests, and this information must be included in the Materials and methods section. It also seems that a serious methodological error is the lack of a control group or the determination of the studied indicators before the pandemic period in the respondents.
111-112: The metabolic equivalent of task (MET) is calculated based on the participant's responses in the questionnaire…. The use of MET determined based on the questionnaire does not seem to be an appropriate method of assessing the actual caloric expenditure, because, as shown by studies calculated during exercise, this indicator is often higher than that resulting from the survey estimates
148-152: All the procedures described above ……and kept a proper distance. I suggest moving to the section of Designed of the study.
Author Response
We would like to thank you for objective and thorough review. Proper comments were taken into account and there were contribute to improve the scientific level of the paper.
40-46: I suggest removing the fragment because it adds little to the substantive part of the manuscript and only makes it less readable
In our opinion, this fragment is important because it indicates the changes that are taking place in the lifestyles of young adults as they enter university. It is reasonable to assume that changes in dietary or physical activity habits will have an impact on students' body composition, which was part of our study. In addition, we recognized that when lockdown was introduced, some students who had to return to their family homes for longer periods of time also returned to their previous habits. Therefore, highlighting these changes seems reasonable to us.
74-77: Given the characteristics of youth……of forced isolation- It’s an assumption which should be removed to the Discussion section or if it is a fact that can be confirmed by the literature on the subject so an adequate reference is needed.
We agree with the Reviewer's comment - we used popular opinion, not scientific sources. We have therefore removed this part from the text.
79-84: In our research we focused on a group particularly predisposed to being physically active: …. towards it. As above- It’s only an assumption that should be removed to the Discussion section or if it is a fact that can be confirmed by the literature on the subject so the adequate reference is needed. Because not every student nor most of them of this field may be interested in being physically active, but only looking for a pleasant job and the opportunity to spend time/work in beautiful and interesting corners of the world, as an animator, not necessarily from physical group activities, but rather as a guide for excursions, for example.
We agree that not every Tourism and Recreation student is physically active. We are also aware that not all graduates will promote active forms of leisure in the future. However, our assumption about the "uniqueness" of this group in terms of physical activity is based on the fact, that it is a field of study in which physical recreation is an important part of the education. The university curriculum provides a wide range of subjects that, both theoretically and practically, prepare students to be specialists in active forms of leisure.
89-96: it is not a Designed of the study, but rather a statement of the fact about the situation in the country during the pandemic period. So it is a suggestion to move this part to the Introduction. The section Design of the Study should contain chapter-specific information.
We agree with this comment and we re-wrote "Design and participants" sub-section.
103-104: All Participants should be described in a more precise way, i.e. height, body weight, BMI. Moreover, in the Materials and Methods section, information about the health of the study participants is stored. Did they suffer from COVID-19, whether they were before or after vaccination, whether they suffered from other ailments, whether they used any specific dietary intervention, whether their condition and diet were similar, whether they supplemented products with an ergogenic effect or before the PVC 170 test? have used CNS stimulants such as caffeine. All these factors affect the results of stress tests, as well as mental well-being and the results of the questionnaire tests, and this information must be included in the Materials and methods section.
Anthropomorphic characteristics of the participants is presented in Table 1. In "Material and Methods" section we provided more information about recruitment for our study. We asked the potential participants about their personal COVID-19 experience. The COVID-19 infection history was an exclusion criterion. We asked the recruited participants not to change their eating habits during the study and we asked them to refrain from alcohol, caffeine and heavy physical effort for 24 hours before the measurements. All this information were added to "Material and Methods" section.
It also seems that a serious methodological error is the lack of a control group or the determination of the studied indicators before the pandemic period in the respondents.
We agree that comparison with control group would be beneficial for this study. On the other hand, conducting the current study in pandemic circumstances was difficult in terms of organizational matters. During on-line education we did not have the face-to-face contact with the students of other faculties, and the students' dormitories were closed. We suppose that students of Tourism and Recreation agreed to participate in our study because they were acquainted with us due to the classes before the pandemic.
111-112: The metabolic equivalent of task (MET) is calculated based on the participant's responses in the questionnaire…. The use of MET determined based on the questionnaire does not seem to be an appropriate method of assessing the actual caloric expenditure, because, as shown by studies calculated during exercise, this indicator is often higher than that resulting from the survey estimates
We agree with this comment and we presented the proper explanation in the last paragraph of "Discussion" section.
148-152: All the procedures described above ……and kept a proper distance. I suggest moving to the section of Designed of the study.
This correction was done.
Reviewer 4 Report
Read the annexed document

Author Response
We would like to thank you for objective and thorough review. Proper comments were taken into account and there were contribute to improve the scientific level of the paper.
- a) Introduction.
- In lines 40 to 46 can be interesting incorporated some information around the importance role of parents in the physical activity of adolescents. I think this estudies are very important and recents, and can be incorporataed. o https://doi.org/10.3390/ijerph18147666 o https://doi.org/10.3390/su11247080
We agree that education for physical activity (by parents and other social institutions) is very important, but this issue was not the subject of our study.
- b) Material and Methods. Some suggestions are proposed.
- Explain more extensive the process of selecting the sample of participants. Explaining if during the process, there has been experimental death of any participant.
This information was added to "Design and participants" sub-section. The number of participants did not changed during the study.
- Explain in more detail the inclusion and exclusion criteria to participate in this study, and ethical consideration.
Information about inclusion and exclusion criteria are now presented in "Design and participants" sub-section. Ethical consideration is presented after "Discussion" section.
- It may be of interest to include a figure showing the flow of participants, all stages of this study.
We are not sure about the meaning of this suggestion.
- Include positive inform from the ethics committee.
The form is attached to correspondence with the Reviewer.
- c) Introduction and discussion: It is recommended to review the discussion and introduction section and incorporate the following recent studies. Thus expanding the citations and incorporating it into the bibliographic references section. In addition to other works that analyze the importance of parental support and the influence of the school environment for the generation of healthy habits. • https://doi.org/10.3390/ijerph16132333
We believe that the issue of parental influence on parenting (including parenting for physical activity) requires a separate study. Our research did not address this issue.
- d) Results: line 166 to 166 appears to show the title of table 2 and line 169 to 17 shows the title again, but slightly modified.
The text introducing Table 2 was modified
- e) Conclusion, limited and future prospects: I think it may be appropriate to do this part of the study with three different contexts: first to explain the conclusion, second to explain what is limited and finally to explain the future prospects.
In our opinion the "Conclusions" section summarizes the main findings of the study. We reviewed many articles already published in IJERPH and we found out that our "Conclusions" are written in similar style to others. The limitations of the study were not presented in any "Conclusions" of other articles. Our "Conclusions" present the future prospects.
- f) References: the 8 reference appears in the red color. The 44 reference appears whit the most biggest letter
Probably the view of manuscript content depends on computer's software. I can see the text in black color and in normal proportions.

Round 2
Reviewer 3 Report
No other comments. Manuscript accepted.
Author Response
No other comments. Manuscript accepted.
Thank you for your acceptance.